# Development of a multidimensional prediction model for long-term prognostic risk in patients with acute coronary syndromes after percutaneous coronary intervention: A retrospective observational cohort study

Bojian Wang(ID)[1], Yanwei Du[1], Pengyu Cao(ID)[2]*, Min Liu[2], Jinting Yang[1], Ningning Zhang[2], Wangshu Shao[3], Lijing Zhao[1], Rongyu Li[3], Lin Wang[3]

1 School of Nursing, Jilin University, Changchun, Jilin, China, 2 The Second People's Hospital of Changzhou, The Third Affiliated Hospital of Nanjing Medical University, Changzhou Jiangsu, China, 3 The Cardiovascular Center, First Hospital of Jilin University, Changchun, Jilin, China

☯ These authors contributed equally to this work.
* caopy1979@outlook.com

## Abstract

### Background

The aim of this study is to examine the critical variables that impact the long-term prognosis of patients with acute coronary syndrome (ACS) after percutaneous coronary intervention (PCI) and to create a multidimensional predictive risk assessment model that can serve as a theoretical basis for accurate cardiac rehabilitation.

### Methods

The study involved ACS patients who received PCI at the First Hospital of Jilin University from June 2020 to March 2021. Participants were categorized into two groups: acute myocardial infarction (AMI) and unstable angina (UA), according to clinical data and angiographic findings. Hospitalization data, physical performance, exercise tolerance prior to discharge, average daily steps, major adverse cardiac events (MACE), and a follow-up period of 36 months were documented. The dates for accessing data for research purposes are February 10, 2022 (10/2/2022) to December 10, 2023 (10/12/2023).

### Results

We observed substantial increases in weight, fasting plasma glucose (FPG), total cholesterol, high-density lipoprotein cholesterol (HDL-C), low-density lipoprotein cholesterol (LDL-C), white blood cell (WBC) count, neutrophil granulocyte count, monocyte count, hemoglobin (Hb) levels, aspartate aminotransferase (AST), and alanine aminotransferase (ALT) levels in the acute myocardial infarction (AMI) cohort

**Data availability statement:** All relevant data are within the paper and its Supporting Information files.

**Funding:** Dr. Cao is supported by Science and Technology Development Plan of Jilin Province: No. 20210204119YY, and Research Fund of Changzhou Medical Center: NO. CZKY102RC202303. The funding bodies were not involved in the study design, data collection or analysis, or writing of the manuscript.

**Competing interests:** The authors have declared that no competing interests exist.

relative to the unstable angina (UA) cohort. We found white blood cell count (WBC) (OR: 4.110) and the effective average number of daily steps (ANS) (OR: 2.689) as independent prognostic risk factors for acute myocardial infarction (AMI). The independent risk factors for unstable angina prognosis were white blood cell count (OR: 6.257), VO2 at anaerobic threshold (OR: 4.294), and effective autonomic nervous system function (OR: 4.097). The whole prognostic risk assessment score for acute myocardial infarction (AMI) is 5 points, with 0 points signifying low risk, 2–3 points representing intermediate risk, and 5 points indicating high risk. The overall prognostic risk assessment score for UA is 7 points, with 0–3 classified as low risk, 4–5 as intermediate risk, and 6–7 as high risk.

## Conclusion

This study developed a multimodal predictive model that integrates the inflammatory response after onset, physical performance and exercise tolerance before discharge, and daily activity after discharge to predict the long-term prognosis of patients with ACS. The multidimensional model is more effective than the single-factor model for assessing risk in ACS patients. This work also establishes a theoretical basis for improving the prognosis of potentially high-risk individuals with accurate and reasonable exercise prescriptions.

## Introduction

Acute coronary syndrome (ACS) is an acute ischemic syndrome of the heart caused by insufficient blood supply to the heart triggered by the rupture of vascular plaques and thrombosis in the coronary arteries [1], which is a major adverse cardiovascular event in clinical practice [2]. Percutaneous coronary intervention (PCI) is the preferred treatment method for patients with ACS, which can quickly unblock blocked vessels, restore myocardial perfusion, and reduce the area of myocardial ischemia [3]. The optimization of clinical management and pharmacological therapy, the in-hospital and 6-month post-discharge mortality rates after PCI in patients with ACS have decreased dramatically [4], however, the long-term prognosis of 1–5 years after the procedure is not optimistic over time. The Global Registry of Acute Coronary Events (GRACE) study showed that the mortality rate in ACS patients was approximately 15% after 1 year and up to 20% cumulative mortality after 5 years [5].

Patients with ACS suffer from myocardial injury, which triggers a decline in physical performance and cardiorespiratory endurance, leading to exercise intolerance and reduced quality of life [6,7], which may be important factors contributing to poor long-term prognosis. The Short Physical Performance Battery (SPPB) is a rapid and objective test of physical performance that has application in the stratification of frailty status, fall risk assessment, and other major adverse health outcomes (including all-cause mortality, disability, and hospital readmission) in older adults [8,9]. However, few studies have been reported on the prognostic assessment of SPPB in patients with

ACS. Cardiopulmonary exercise test (CPET) and six-minute walk test (6MWT) are two noninvasive methods for quantitatively assessing cardiac reserve function and exercise tolerance, which are often used in clinical practice for diagnosis and prognostic assessment of cardiopulmonary diseases and to characterize subjects' exercise tolerance [10–12]. Our previous studies have found that oxygen consumption <10.5ml/min/kg at the anaerobic threshold (AT), a parameter of cardiopulmonary exercise stress testing, is an independent risk factor for poor prognosis in patients with AMI [13].

Exercise-based cardiac rehabilitation could not only improve exercise capacity and quality of life but also impose a positive effect on the prognoses of patients with ACS [14]. Walking is the safest and most common form of exercise rehabilitation for ACS patients after discharge from the hospital, and the amount of exercise can be reflected by the number of steps recorded by portable devices. Relevant studies have used daily steps as a simple and effective measure of daily activity to investigate the relationship between cardiovascular disease and daily activity [15,16]. Recent studies have shown that a high number of daily steps in the general population does not additionally reduce the incidence of cardiovascular events and all-cause mortality and that the optimal number of steps to reduce the incidence of cardiovascular events in people over 60 years of age is approximately 7,200 steps [17]. However, there are few studies on the correlation between the number of daily steps and prognosis after PCI in ACS patients.

To our knowledge, few prognostic prediction models for ACS patients incorporate exercise endurance and daily activity. Traditional prognostic assessment of ACS patients has focused on age, gender, history, basic clinical biochemical indices after the onset of the disease (triglyceride-glucose index (TyG index), platelet-to-lymphocyte ratio (PLR)), as well as static cardiac function indices (left ventricular ejection fraction) [18–20]. In recent years, more and more studies have focused on the prognostic impact of factors related to the level of exercise cardiorespiratory endurance in ACS patients after surgery [13], and daily physical activity after discharge [21,22]. However, the predictive assessment of ACS patients after surgery lacks a multidimensional and comprehensive model for the consideration of the clinical biochemical indexes after the onset of the disease, the indexes of exercise endurance before hospital discharge, and the indexes of daily physical activity after hospital discharge, etc.; thus, individualized dynamics risk assessment to improve long-term prognosis cannot be performed. It is also not possible to determine the extent to which these three aspects have an impact on the prognosis of the patient.

This retrospective cohort study looked at the predictive effects of clinical biochemical indexes, physical functional status, exercise cardiorespiratory endurance level, and daily exercise over three time periods: after onset, before discharge, and after discharge. Furthermore, we developed a multidimensional joint effect model with numerous independent risk factors to assist in identifying high-risk patients and making tailored therapeutic decisions for cardiac rehabilitation and secondary prevention.

## Methods

### Study design and ethical issues

This retrospective cohort study was conducted from February 10, 2022 (10/2/2022) to December 10, 2023 (10/12/2023) at the First Hospital of Jilin University, China. The study data were collected from patients who were diagnosed with ACS and underwent PCI in the First Hospital of Jilin University from June 1, 2020 to March 1, 2021。 All procedures in this study were performed according to the principles of the Declaration of Helsinki. The Medical Ethics Committee of the First Hospital of Jilin University approved this study (Approval No. AF-IRB-032-06, AF-IRB-029-06). The study was registered with the Chinese Clinical Trials Register (ChiCTR2300068294) [23]. The registration date is February 14, 2023. The dates for accessing data for research purposes are February 10, 2022 (10/2/2022) to December 10, 2023 (10/2/2022).

### Subjects

From June 1, 2020, to March 31, 2021, we collected data from 1,899 patients presenting with ACS and undergoing PCI at the First Hospital of Jilin University. The study protocol received approval from the institutional review board of the First Hospital of Jilin University (Approval No. AF-IRB-032-06). Before they participated in the study, all participants provided

written informed consent. According to the 2020 ESC Guidelines for the Management of acute coronary syndromes (ACS) [24], unstable angina (UA) was defined as myocardial ischemia at rest or on minimal exertion in the absence of acute cardiomyocyte injury/necrosis. Among the participants with unstable angina, coronary angiography revealed at least one vessel with a stenosis greater than 50%. Acute myocardial infarction (AMI) was characterized by severe chest pain, troponin elevation, and evidence of vascular stenosis or blockage on coronary angiography. Additionally, various data were obtained from the patient's medical records, including age, sex, height, weight, history of hypertension, history of diabetes, history of hyperlipidemia, triglyceride levels, total cholesterol levels, low-density lipoprotein cholesterol (LDL-C) levels, high-density lipoprotein cholesterol (HDL-C) levels, fasting plasma glucose (FPG) levels, aspartate aminotransferase (AST) levels, alanine aminotransferase (ALT) levels, creatinine levels, urea nitrogen levels, uric acid levels, hemoglobin (Hb) levels, white blood cell (WBC) counts, lymphocyte counts, neutrophil granulocyte counts, monocyte counts, platelet counts, Natrium levels, potassium levels, ejection fraction (EF), the end-diastolic diameter of the left ventricle (EDLV), lesion vessel information, and the number of stents.

## Physical performance

On the day of discharge, physical performance was evaluated using the SPPB [8], which consists of three timed components: three balance stances, a 4-meter usual-pace walk, and five repeated chair stands. Component times were converted to scores ranging from 0 to 4, with higher scores indicating greater functional capacity. A score of 0 was given if the participant found the task too risky or was unable to complete it. The three component scores were then summed to create a total score ranging from 0 to 12.

## Exercise tolerance

To accurately quantify exercise tolerance, we used cardiopulmonary exercise testing (CPX) for UA, and 6 min walking distance test (6MWT) for AMI on the day of discharge, which are both the widely accepted evaluation tools in both the United States (US), Europe and China[10–12]. In the UA patients, oxygen consumption (VO2) at anaerobic threshold (AT) was measured in standard exercise testing using Cardio-respiratory instrumentation Medisoft (MS, made in Belgium, SN:130619–05-1470, Model: E100000011000001). The exercise tolerance was estimated from the bicycle cycle ergometer work rate. The use of CPX during progressive exercise (10 watts per minute) is based on the measurement of exercise gas exchange. The exercise test was terminated if any of the following occurred: abnormal hemodynamic or ECG exercise response or other reasons (i.e., lower extremity muscle fatigue, angina, and dyspnoea). The exercise load was determined by a cycle ergometer (Ergoselect 100P, ergoline GmbH, Germany) work rate. The progressive load was 10 watts per minute during the graded exercise test, and the pedaling cadence was kept at 55–65 revolutions per minute (RPM) throughout the test [12]. In the AMI patients, 6MWT was performed according to the guidelines set forth by the American Thoracic Society [11,25]. The patients were instructed to walk as much as possible in 6 min along a 30 m corridor while continuously monitoring their oxygen saturation and heart rate. The 6MWT was discontinued if (1) the SpO$_2$ remained below 90%, (2) if the HR was > 130, < 50 bpm, or increased to ≥ 30 bpm before the task, (3) upon the occurrence of a new arrhythmia, (4) upon the occurrence of subjective symptoms (dizziness, nausea, chest pain, headache, intense fatigue, cold sensation, cold sweat, and marked dyspnea), and (5) when safe monitoring could not be performed. The outcomes during implementation were the total 6 min walking distance (6MWD).

## Average number of steps

After discharge, the patient's daily step count within one year was obtained through the mobile communication APP software (Wechat). Daily steps ≥ 1,000 are considered the effective daily number of steps, and the effective daily average number of steps (ANS) throughout the year was recorded. The monthly ANS is the ratio of the total number of steps to the total number of months in a year.

## Clinical follow-up

Follow-up data was acquired through hospital records and telephone interviews which were conducted every 3 months from discharge until cardiac death or December 31, 2023, whichever came first. Major adverse cardiac events (MACE) including cardiogenic death and re-hospitalization such as heart failure, stroke, and ACS were documented. Patients with cardiac death who lost telephone interviews were identified from the population registry bureau. The average duration of follow-up was 3 years.

## Statistical analysis

Continuous variables following a normal distribution were described as mean ± standard deviation, while non-normally distributed variables were presented as medians with interquartile range (IQR). Categorical data were reported as percentages and raw numbers. Continuous variable statistical comparisons between groups were performed using the t-test if normally distributed, and the Mann-Whitney U test if non-normally distributed. The chi-square or Fisher's exact test was used for categorical data. The univariate analyses were adjusted for age, sex, height, and weight including variables with a p-value <0.05. The Multivariable logistic regression was used to identify independent risk factors for prognosis. The predictive value of the model for MACE was assessed using the receiver operating characteristic (ROC) curve. Correlation analysis was conducted using the Pearson test, and survival analysis was performed using the Kaplan-Meier curve. All statistical analyses were conducted using SPSS 23.0 software (IBM Corp., Armonk, NY, USA), and a two-tailed $P < 0.05$ was considered statistically significant.

# Results

## Participants characteristics

Between June 2020 and March 2021, 1899 patients were diagnosed with ACS and underwent PCI at the First Hospital of Jilin University. Patients without a daily walking record (n = 1004), patients with a walking record of fewer than 6 months (n = 459), and patients without data on SPPB, 6MWT, and CEPT before discharge (n = 117) were excluded. The remaining patients (n = 319) were included in the follow-up. Ultimately, 287 patients were enrolled in this study, with 32 patients lost to follow-up, the loss rate was 9.7%. Based on the clinical diagnosis, the participants were divided into two groups: the AMI group and the UA group. The AMI group consisted of 185 participants (154 in the NO-MACE group and 31 in the MACE group), and the UA group consisted of 102 participants (82 in the NO-MACE group and 20 in the MACE group) (Fig 1).

The mean age of the AMI group was 53.56, 85.95% were male, and the mean age of the UA group was 56.01, 80.39% were male. Weight, FPG, Total cholesterol, HDL-C, LDL-C, WBC, Neutrophile granulocyte, Monocyte, Hb, AST, and ALT were significantly higher, while Age and EF were significantly lower in the AMI group compared to the UA group (Table 1). These findings propose that individuals with AMI and UA exhibit distinct pathophysiological states. Consequently, it implies a nuanced requirement for tailored prognostic prediction models to effectively discern and anticipate the diverse clinical outcomes associated with each condition.

## Screening prediction variable

We conducted a comprehensive univariate screening involving 37 variables at the preoperative, pre-discharge, and post-discharge stages. In the AMI patients, the MACE group exhibited significantly lower levels of WBC, Neutrophilic Granulocytes, and Monocytes compared to the No-MACE group; moreover, the 5STS, monthly ANS, and effective ANS were markedly higher in the MACE group than the No-MACE group (Table 2). In the UA patients, the MACE group showed significantly lower levels of WBC, Neutrophile granulocyte, Monocyte, SPPB score, $VO_2$ at AT, ANS and Effective ANS than the No-MACE group (Table 2).

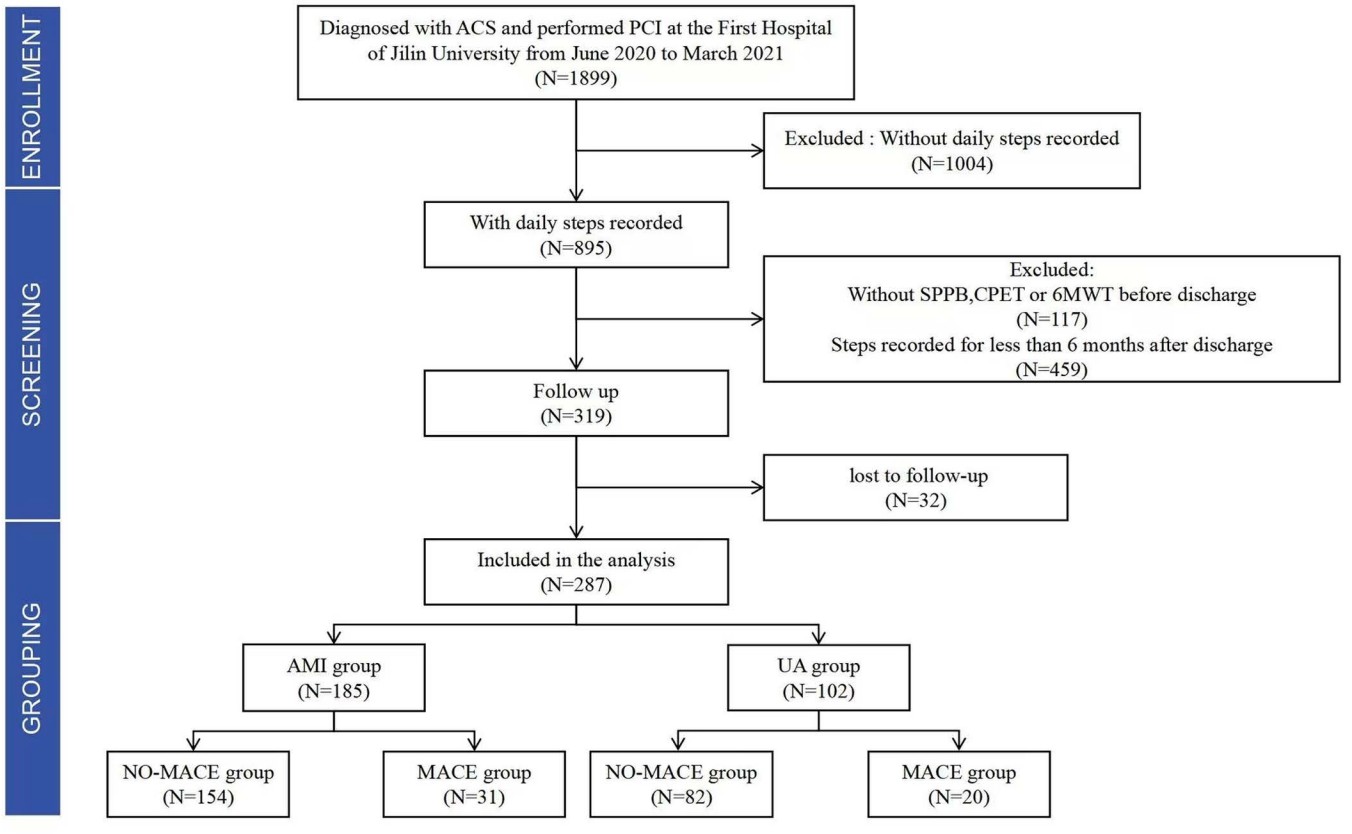

**Fig 1. Flowchart from enrollment to grouping.**

Given that Neutrophilic Granulocytes and Monocytes were components of WBC counts, and ANS and Effective ANS were derived through similar calculations, correlation analyses were conducted for these two sets of variables. In the AMI group, significant linear correlations were observed between WBC and Neutrophilic Granulocytes ($y = 0.9252x + 3.003$, $R2 = 0.8135$, $P < 0.0001$), as well as WBC and Monocytes ($y = 4.016x + 6.131$, $R2 = 0.1523$, $P < 0.0001$). ANS exhibited a pronounced linear correlation with Effective ANS ($y = 29.55x - 26883$, $R2 = 0.9404$, $P < 0.0001$). In the UA group, significant linear correlations were observed between WBC and Neutrophilic Granulocytes ($y = 0.6488x + 4.491$, $R2 = 0.3871$, $P < 0.0001$), WBC and Monocytes ($y = 5.390x + 4.489$, $R2 = 0.2960$, $P < 0.0001$), as well as ANS and Effective ANS ($y = 30.27x - 32727$, $P < 0.0001$).

The ROC curves were generated to assess the predictive efficacy of each aforementioned factor for MACE. The cut-off values for the AMI group were WBC ($6.46 \times 109/L$, AUC = 0.641, Sensitivity = 0.419, Specificity = 0.850), 5STS (19.47s, AUC = 0.617, Sensitivity = 0.552, Specificity = 0.683), effective ANS (6093 steps/day, AUC = 0.620, Sensitivity = 0.806, Specificity = 0.429) (Fig 2). The cut-off values for the UA group were WBC ($6.44 \times 109/L$, AUC = 0.685, Sensitivity = 0.737, Specificity = 0.679), SPPB score (9.50, AUC = 0.639, Sensitivity = 0.850, Specificity = 0.451), $VO_2$ at AT (2.95Mets, AUC = 0.675, Sensitivity = 0.700, Specificity = 0.630), effective ANS (5841steps/day, AUC = 0.701, Sensitivity = 0.650, Specificity = 0.768) (Fig 3).

Ultimately, WBC, 5STS, and effective ANS were chosen for inclusion in the logistic regression analysis for the AMI group. Similarly, WBC, SPPB score, $VO_2$ at AT, and effective ANS were selected for the UA group in the logistic regression analysis. One-way logistic regression analyses, adjusting for confounders including age, sex, height, and weight, revealed

**Table 1. Comparison of clinical data and baseline data between AMI and UA.**

| | AMI (n = 185) | UA (n = 102) | P |
|---|---|---|---|
| Age (years) | 53.56 ± 9.71 | 56.01 ± 8.19 | **0.024** |
| Sex, male, n (%) | 159 (85.95) | 82 (80.39) | 0.220 |
| High (m), (IQR) | 1.70 (1.65, 1.75) | 1.69 (1.64, 1.72) | 0.077 |
| Weight (kg), (IQR) | 77.00 (65.50, 83.00) | 71.00 (65.00, 77.25) | **0.017** |
| BMI (kg/m$^2$) | 26.11 ± 3.45 | 26.11 ± 3.26 | 0.092 |
| History of hypertension, n (%) | 83 (44.86) | 46 (45.10) | 0.970 |
| History of diabetes, n (%) | 47 (25.41) | 26 (25.49) | 0.987 |
| History of hyperlipidemia, n (%) | 136 (73.51) | 68 (66.67) | 0.221 |
| EF (%), (IQR) | 57.00 (55.00, 60.00) | 60.00 (60.00, 63.00) | **<0.001** |
| EDLV (mm), (IQR) | 50.00 (47.00, 52.75) | 48.00 (45.00, 53.00) | 1.000 |
| FPG (mmol/L), (IQR) | 6.32 (5.38, 7.82) | 5.77 (5.135, 6.83) | **0.025** |
| Triglyceride (mmol/L), (IQR) | 1.84 (1.27, 2.57) | 1.63 (1.14, 2.31) | 0.114 |
| Total cholesterol (mmol/L) | 4.82 ± 1.15 | 4.11 ± 1.00 | **<0.001** |
| HDL-C (mmol/l) | 1.02 ± 0.22 | 0.97 ± 0.18 | **0.035** |
| LDL-C (mmol/l) | 3.11 ± 0.92 | 2.57 ± 0.78 | **<0.001** |
| TyG, (IQR) | 4.92 (4.71, 5.17) | 4.86 (4.86, 5.09) | **0.065** |
| WBC (10$^9$/L), (IQR) | 8.33 (6.83, 10.05) | 6.95 (5.98, 8.50) | **<0.001** |
| Lymphocyte | 1.77 (1.41, 2.16) | 1.83 (1.44, 2.16) | 0.396 |
| Neutrophile granulocyte | 5.49 (4.24, 7.30) | 4.23 (3.49, 5.14) | **<0.001** |
| Monocyte | 0.59 (0.43, 0.74) | 0.47 (0.39, 0.59) | **<0.001** |
| Hb (g/L), (IQR) | 150.00 (138.00, 159.50) | 143.00 (136.00, 153.75) | **0.017** |
| Platelet (10$^9$/L), (IQR) | 220.00 (189.00, 252.50) | 224.50 (188.00, 266.25) | 0.681 |
| AST (U/L), (IQR) | 53.75 (27.65, 107.55) | 18.70 (15.73, 24.28) | **<0.001** |
| ALT (U/L), (IQR) | 28.85 (21.20, 48.48) | 21.75 (15.70, 32.00) | **<0.001** |
| Urea nitrogen (mmol/L), (IQR) | 5.32 (4.52, 6.33) | 5.54 (4.78, 6.51) | 0.325 |
| Uric acid (mmol/l), (IQR) | 352.00 (292.00, 405.00) | 347.00 (312.00, 407.50) | 0.731 |
| Creatinine (mmol/L), (IQR) | 67.10 (59.90, 78.95) | 71.60 (62.35, 81.98) | 0.105 |
| Hb (g/dL)/Cr (mg/dL) (IQR) | 28.71 (23.89, 35.55) | 18.09 (15.54, 20.74) | **<0.001** |
| Natrium (mmol/L) | 138.28 ± 2.49 | 139.02 ± 2.34 | 0.160 |
| Potassium (mmol/L) | 3.98 ± 0.35 | 4.04 ± 0.31 | 0.185 |
| SPPB (score), (IQR)† | 9.00 (8.00, 10.00) | 9.00 (9.00, 10.00) | 0.183 |
| Standing balance (score) | 4.00 (4.00, 4.00) | 4.00 (4.00, 4.00) | 0.207 |
| 4MGS (s) | 4.46 (3.94, 5.11) | 4.36 (3.87, 4.86) | 0.147 |
| 5STS (s) | 17.22 (14.51, 21.36) | 16.90 (13.89, 19.87) | 0.153 |
| ANS (steps/month) | 183531.13 ± 81207.35 | 188238.80 ± 72804.75 | 0.628 |
| Effective ANS (steps/day) | 7121.68 ± 2665.32 | 7300.27 ± 2379.59 | 0.573 |

Note: AMI vs. UA (p-value); Significant probabilities were marked in bold.

BMI: body mass index, EF: Ejection fraction, EDLV: End diastolic diameter of left ventricle, FPG: fasting plasma glucose, HDL-C: High density lipoprotein cholesterol, LDL-C: Low density lipoprotein cholesterol, TyG: triglyceride glucose index WBC: white blood cell, Hb: hemoglobin, AST: Aspartate amino-transferase, ALT: Alanine aminotransferase, SPPB: Short Physical Performance Battery, 4MGS: 4-meter gait-speed, 5STS: 5-repetition Sit-To-Stand, VO$_2$ at AT: Oxygen consumption per kilogram of weight per minute at anaerobic threshold, ANS: Average number of steps.

†This analysis included data from 178 patients in the AMI group and 102 patients in the UA group.

**Table 2. Comparison of clinical data and baseline data between AMI and UA.**

| | AMI (n = 185) | | | UA (n = 102) | | |
|---|---|---|---|---|---|---|
| | Non-MACE (n = 154) | MACE (n = 31) | *P* | Non-MACE (n = 82) | MACE (n = 20) | *P* |
| Age (years) | 53.64 ± 9.53 | 53.13 ± 10.70 | 0.789 | 55.77 ± 7.89 | 57.00 ± 9.45 | 0.549 |
| Sex, male, n (%) | 132 (85.70) | 27 (87.1) | 1.000 | 67 (81.70) | 15 (75.00) | 0.716 |
| High (m), (IQR) | 1.70 (1.64, 1.75) | 1.70 (1.67, 1.75) | 0.504 | 1.69 (1.64, 1.72) | 1.69 (1.61, 1.70) | 0.582 |
| Weight (kg), (IQR) | 77.00 (66.75, 83.00) | 75.00 (64.0, 85.00) | 0.884 | 71.00 (65.00, 76.00) | 71.50 (63.50, 80.00) | 0.606 |
| BMI (kg/m²), (IQR) | 26.16 ± 3.16 | 25.86 ± 3.75 | 0.643 | 25.27 ± 3.51 | 26.04 ± 3.21 | 0.374 |
| History of hypertension, n (%) | 70 (45.50) | 13 (41.90) | 0.719 | 37 (45.10) | 9 (45.00) | 0.992 |
| History of diabetes, n (%) | 42 (27.30) | 5 (16.10) | 0.193 | 21 (25.60) | 5 (25.00) | 0.955 |
| History of hyperlipidemia, n (%) | 114 (74.03) | 22 (70.97) | 0.280 | 52 (63.41) | 16 (80.00) | 0.158 |
| Killip, n (%) | | | 0.760 | | | |
| I | 133 (86.36) | 28 (90.32) | | | | |
| II or III | 21 (13.63) | 3 (9.67) | | | | |
| EF (%), (IQR) | 57.00 (55.00, 60.00) | 58.00 (53.00, 60.00) | 0.745 | 60.00 (59.50, 63.00) | 62.00 (60.00, 63.00) | 0.497 |
| EF < 50%, n (%) | 16 (10.39) | 6 (19.35) | 0.270 | | | |
| EDLV (mm), (IQR) | 50.00 (47.00, 53.00) | 50.00 (46.00, 52.00) | 0.692 | 48.00 (45.00, 50.00) | 49.00 (47.00, 50.50) | 0.237 |
| ST elevation, n (%) | 100 (64.64) | 21 (67.74) | 0.764 | | | |
| Troponin (ng/ml), (IQR) | 2.03 (0.36, 6.28) | 1.82 (0.23, 6.36) | 0.257 | | | |
| FPG (mmol/L), (IQR) | 6.42 (5.43, 8.05) | 6.06 (5.13, 7.60) | 0.282 | 5.77 (5.16, 6.96) | 5.77 (5.03, 6.84) | 0.676 |
| Triglyceride (mmol/L), (IQR) | 1.84 (1.28, 2.74) | 1.87 (1.26, 2.18) | 0.475 | 1.70 (1.14, 2.28) | 1.49 (1.06, 2.39) | 0.421 |
| Total cholesterol (mmol/L) | 4.84 ± 1.14 | 4.72 ± 1.23 | 0.579 | 4.19 ± 0.97 | 3.79 ± 1.05 | 0.124 |
| HDL-C (mmol/l) | 1.03 ± 0.22 | 1.00 ± 0.23 | 0.545 | 0.97 ± 0.18 | 0.97 ± 0.19 | 0.877 |
| LDL-C (mmol/l) | 3.14 ± 0.91 | 2.96 ± 0.92 | 0.303 | 2.63 ± 0.75 | 2.32 ± 0.90 | 0.124 |
| TyG, (IQR) | 4.92 (4.72, 5.23) | 4.82 (4.65, 5.09) | 0.288 | 4.89 ± 0.29 | 4.83 ± 0.32 | 0.408 |
| WBC (10⁹/L), (IQR) | 8.77 ± 2.40 | 7.55 ± 2.50 | **0.012** | 7.32 (6.11, 9.01) | 6.03 (5.59, 7.17) | **0.013** |
| Lymphocyte | 1.78 (1.44, 2.16) | 1.65 (1.18, 2.23) | 0.335 | 1.84 (1.44, 2.18) | 1.79 (1.25, 2.36) | 0.528 |
| Neutrophile granulocyte | 5.51 (4.45, 7.51) | 5.14 (3.37, 6.03) | **0.024** | 4.30 (3.54, 5.30) | 3.66 (3.06, 4.33) | **0.017** |
| Monocyte | 0.61 (0.45, 0.77) | 0.47 (0.36, 0.63) | **0.010** | 0.50 (0.42, 0.62) | 0.39 (0.35, 0.55) | **0.045** |
| Hb (g/L), (IQR) | 150.00 (140.00, 159.25) | 142.00 (134.00, 161.00) | 0.249 | 144.80 ± 15.48 | 142.95 ± 11.93 | 0.626 |
| Platelet (10⁹/L), (IQR) | 222.00 (189.75, 253.75) | 215.00 (184.00, 230.00) | 0.427 | 232.25 ± 59.22 | 222.89 ± 57.04 | 0.534 |
| AST (U/L), (IQR) | 54.10 (29.10, 99.70) | 53.40 (24.70, 156.40) | 0.766 | 18.65 (15.79, 23.13) | 20.25 (16.75, 20.43) | 0.648 |
| ALT (U/L), (IQR) | 28.30 (21.50, 47.90) | 32.00 (20.90, 49.70) | 0.637 | 21.95 (16.05, 31.38) | 19.95 (13.15, 41.90) | 0.077 |
| Urea nitrogen (mmol/L), (IQR) | 5.23 (4.49, 6.27) | 5.73 (4.96, 6.96) | 0.112 | 5.58 ± 1.19 | 5.55 ± 1.24 | 0.907 |
| Uric acid (mmol/l), (IQR) | 356.00 (291.50, 406.50) | 344.00 (304.00, 405.00) | 0.575 | 367.52 ± 92.88 | 333.95 ± 78.53 | 0.149 |
| Creatinine (mmol/L), (IQR) | 67.20 (59.90, 78.43) | 66.60 (59.80, 86.80) | 0.641 | 73.02 ± 14.02 | 68.38 ± 14.47 | 0.200 |
| Hb (g/dL)/Cr (mg/dL) (IQR) | 2.03 (0.36, 6.28) | 28.40 (22.35, 35.55) | 0.530 | 18.09 ± 3.49 | 19.70 ± 4.30 | 0.081 |
| Natrium (mmol/L) | 138.20 ± 2.53 | 138.68 ± 2.28 | 0.326 | 139.07 ± 2.23 | 138.81 ± 2.76 | 0.670 |
| Potassium (mmol/L) | 3.98 ± 0.35 | 3.95 ± 0.36 | 0.698 | 4.04 ± 0.31 | 4.00 ± 0.32 | 0.507 |
| Lesion vessel, n (%) | | | 0.850 | | | 0.689 |
| LM | 2 (1.30) | 0 (0.00) | | 0 (0.00) | 0 (0.00) | |
| LAD | 70 (45.45) | 16 (51.61) | | 45 (54.88) | 12 (60.00) | |
| LCX | 19 (12.34) | 3 (9.68) | | 13 (15.85) | 4 (20.00) | |
| RCA | 63 (40.91) | 12 (38.71) | | 24 (29.27) | 4 (20.00) | |
| No. of stents, n (%) | | | 0.257 | | | 0.252 |
| 1 | 94 (61.04) | 18 (58.06%) | | 41 (50.00%) | 14 (70.00%) | |
| 2 | 37 (24.03) | 11 (35.49%) | | 23 (28.05%) | 4 (20.00%) | |
| ≥3 | 23 (14.93) | 2 (6.45%) | | 18 (21.95%) | 2 (10.00%) | |

*(Continued)*

**Table 2.** (Continued)

| | AMI (n = 185) | | | UA (n = 102) | | |
|---|---|---|---|---|---|---|
| | Non-MACE (n = 154) | MACE (n = 31) | P | Non-MACE (n = 82) | MACE (n = 20) | P |
| SPPB (score), (IQR)∮ | 9.00 (8.00, 10.00) | 9.00 (8.00, 10.00) | 0.268 | 9.00 (9.00, 10.00) | 8.00 (9.00, 9.00) | **0.033** |
| Standing balance (score) | 4.00 (4.00, 4.00) | 4.00 (4.00, 4.00) | 0.850 | 4.00 (4.00, 4.00) | 4.00 (4.00, 4.00) | 0.621 |
| 4MGS (s) | 4.46 (3.85, 5.13) | 4.40 (4.13, 4.91) | 0.880 | 4.32 (3.84, 4.76) | 4.43 (3.97, 4.94) | 0.402 |
| 5STS (s) | 17.09 (14.45, 20.65) | 19.79 (14.96, 24.41) | **0.048** | 16.56 (13.78, 19.42) | 18.96 (15.31, 22.42) | 0.115 |
| 6MWD (m) for AMI ¶ | 279.59 ± 58.57 | 276.76 ± 40.24 | 0.850 | | | |
| VO₂ at AT (Mets) for UA, (IQR) | | | | 3.20 (2.80, 3.80) | 2.85 (2.50, 3.20) | **0.016** |
| Effective ANS (steps/day) | 6923.42 ± 2599.68 | 8233.34 ± 2802.44 | **0.041** | 7614.80 ± 2326.80 | 6010.72 ± 2200.50 | **0.006** |
| ANS (steps/month) | 177462.78 ± 78946.65 | 217557.21 ± 86694.89 | **0.038** | 198403.64 ± 70485.62 | 146562.95 ± 74170.92 | **0.004** |

Note: MACE vs. Non-MACE (p-value); Significant probabilities were marked in bold.

BMI: body mass index, EF: Ejection fraction, EDLV: End diastolic diameter of left ventricle, FPG: Fasting plasma glucose, HDL-C: High density lipoprotein cholesterol, LDL-C: Low density lipoprotein cholesterol, WBC: white blood cell, Hb: hemoglobin, TyG: triglyceride glucose index AST: Aspartate aminotransferase, ALT: Alanine aminotransferase, LM: Left main coronary artery, LAD: Left anterior descending branch, LCX: Left circumflex branch, RCA: Right coronary artery, SPPB: Short Physical Performance Battery, 4MGS: 4-meter gait-speed, 5STS: 5-repetition Sit-To-Stand, 6MWD: 6 min walking distance, VO₂ at AT: Oxygen consumption per kilogram of weight per minute at anaerobic threshold, ANS: Average number of steps.

∮This analysis of the AMI group included data from 149 patients in the NO-MACE group and 29 patients in the MACE group.

¶This analysis of the AMI group included data from 101 patients in the NO-MACE group and 23 patients in the MACE group.

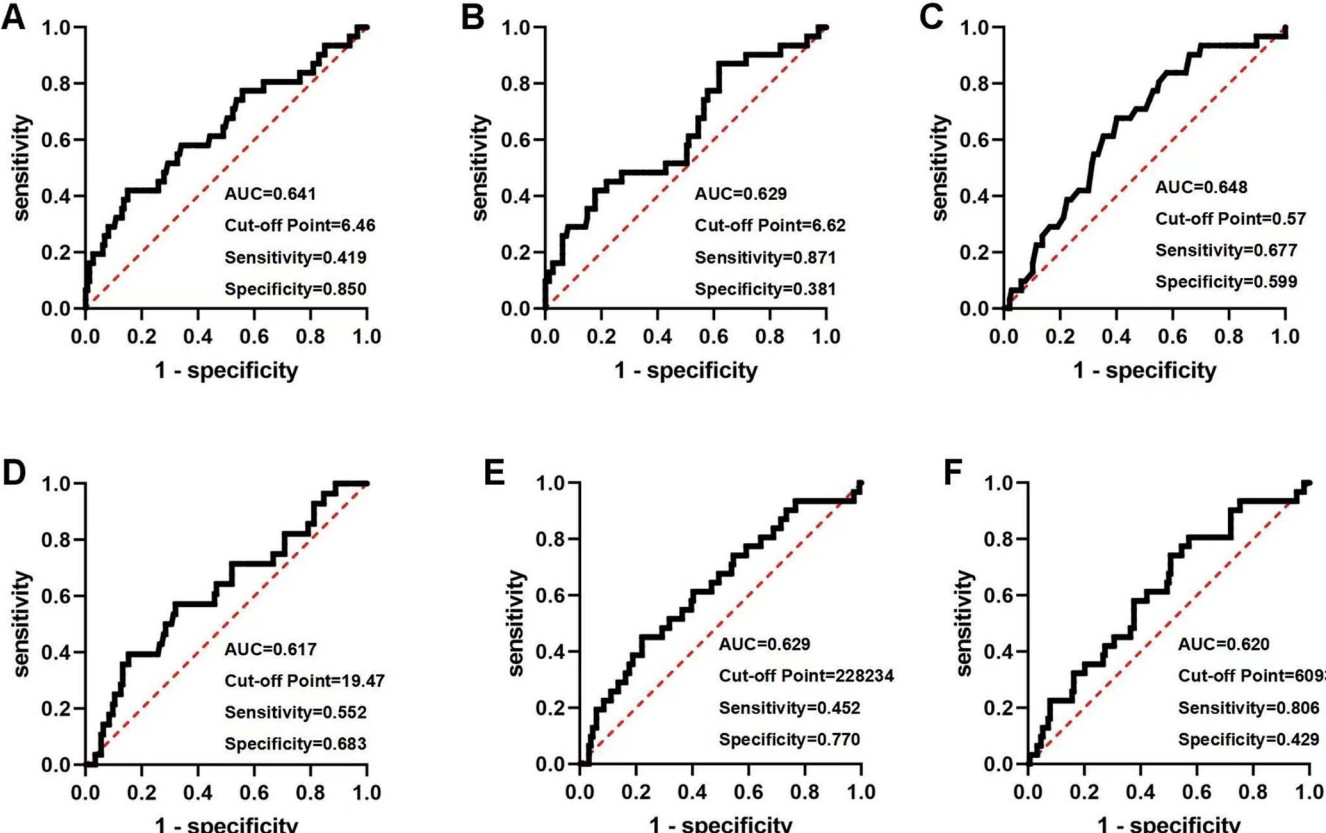

**Fig 2. The ROC curves of AMI patients.** A: White blood cell (WBC); B: Neutrophile granulocyte; C: Monocyte; D: 5-repetition Sit-To-Stand (5STS); E: Average number of steps (ANS); F: Effective ANS.

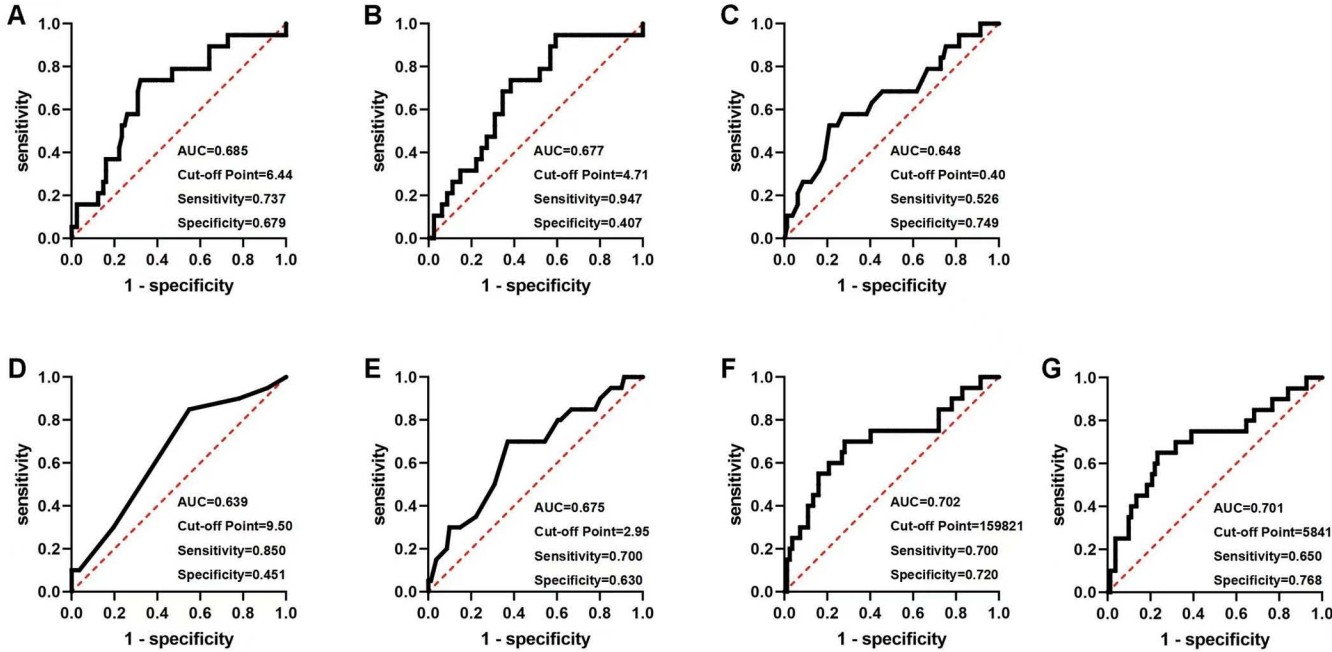

**Fig 3. The ROC curves of UA patients.** A: White blood cell (WBC); B: Neutrophile granulocyte; C: Monocyte; D: Short Physical Performance Battery (SPPB); E: Oxygen consumption per kilogram of weight per minute at anaerobic threshold (VO$_2$ at AT); F: Average number of steps (ANS); G: Effective ANS.

that in the AMI group, WBC (OR = 34.543, P = 0.001) and Effective ANS (OR = 3.239, P = 0.031) exhibited statistical significance (P < 0.05) (Table 3a). Similarly, in the UA group, WBC (OR = 46.346, P = 0.001), SPPB score (OR = 7.115, P = 0.014), VO$_2$ at AT (OR = 4.296, P = 0.012), and Effective ANS (OR = 5.814, P = 0.003) exhibited statistical significance (P < 0.05) (Table 3a).

### Building predictive models

Significantly associated variables, identified through one-way logistic regression and adjusted for confounders, were included in a multifactorial logistic regression analysis. In the AMI group, employing the "Enter" method for screening, WBC (OR = 4.110, P = 0.001) and effective ANS (OR = 2.689, P = 0.046) emerged as independent risk factors influencing the prognosis of AMI. These variables were scored based on the regression coefficient, resulting in a total score of 5, with 3 assigned to WBC and 2 to Effective ANS (Table 3b).

In the UA group, due to the extensive array of variables, a screening approach involving both "Forward" or "Backward" methods was employed. SPPB was excluded from the model. Notably, WBC (OR = 6.257, P = 0.003)), VO$_2$ at AT (OR = 4.294, P = 0.017), and Effective ANS (OR = 4.097, P = 0.018) emerged as independent risk factors influencing the prognosis of UA. These variables were scored based on the regression coefficient, resulting in a total score of 7, with 3 assigned to WBC, 2 to VO$_2$ at AT, and 2 to Effective ANS (Table 3b).

### Model validation and risk stratification

Patients were stratified based on the developed model, and the Kaplan-Meier (K-M) curves were generated according to the timing of postoperative MACE. In the AMI model, the group with a score of 2 exhibited a significantly higher proportion of MACE at 24 and 36 months compared to the combined group of 0. The group with a score of

**Table 3. Logistic regression.**

**a. Univariate logistic regression adjusted for age, sex, height, and weight**

| | Cut-off Point | Regression coefficient | P | OR | 95 percent CI for OR |
|---|---|---|---|---|---|
| **AMI** | | | | | |
| WBC ($10^9$/L) | 6.46 | 1.494 | **0.001** | 4.543 | 1.037-13.336 |
| Effective ANS (steps/day) | 6000 | 1.175 | **0.031** | 3.239 | 1.114-9.419 |
| 5STS | 19.47 | 0.868 | 0.270 | 2.381 | 0.510-11.121 |
| **UA** | | | | | |
| WBC ($10^9$/L) | 6.44 | 1.848 | **0.001** | 6.346 | 2.043-19.706 |
| SPPB (score) | 10 | 1.962 | **0.014** | 7.115 | 1.488-34.013 |
| $VO_2$ at A T (Mets) | 3 | 1.458 | **0.012** | 4.296 | 1.369-13.481 |
| Effective ANS (steps/day) | 6000 | 1.760 | **0.003** | 5.814 | 1.790-18.879 |

**b. Multivariable logistic regression.**

| | Cut-off Point | Regression coefficient | P | OR | 95 percent CI for OR | Categories | Points |
|---|---|---|---|---|---|---|---|
| **AMI (Method: Enter)** | | | | | | | |
| WBC ($10^9$/L) | 6.46 | 1.413 | **0.001** | 4.110 | 1.742-9.695 | ≥6.46 | 0 |
| | | | | | | <6.46 | 3 |
| Effective ANS (steps/day) | 6000 | 0.989 | **0.046** | 2.689 | 1.019-7.095 | ≤6000 | 0 |
| | | | | | | >6000 | 2 |
| **UA (Method: Enter. SPPB was excluded from the model.)** | | | | | | | |
| WBC ($10^9$/L) | 6.44 | 1.834 | **0.003** | 6.257 | 1.837-21.312 | ≥6.44 | 0 |
| | | | | | | <6.44 | 3 |
| $VO_2$ at AT (Mets) | 3 | 1.457 | **0.017** | 4.294 | 1.292-14.269 | ≥3 | 0 |
| | | | | | | <3 | 2 |
| Effective ANS (steps/day) | 6000 | 1.410 | **0.018** | 4.097 | 1.276-13.161 | ≥6000 | 0 |
| | | | | | | <6000 | 2 |

Note: Significant probabilities were marked in bold.

SPPB: Short Physical Performance Battery, 5STS: Five-repetition Sit-To-Stand, $VO_2$ at AT: Oxygen consumption per kilogram of weight per minute at anaerobic threshold, ANS: Average number of steps.

3 demonstrated a significantly elevated proportion of MACE at 12, 24, and 36 months compared to the group of 0. Furthermore, the 5-point population demonstrated significantly elevated proportions of MACE at 24 and 48 months compared to the 0-point population, and at 48 months compared to the 3-point population. While, no significant differences were observed in the occurrence of MACE at 12, 24, and 36 months between the 2 points and 3 points groups (Fig 4A).

In the UA models, the populations with 0, 2, and 3 points were amalgamated, as no significant differences were observed in the occurrence of MACE at 12, 24, and 36 months among these groups. Individuals with a score of 4 showed a markedly higher proportion of MACE at 48 months in contrast to the 0–3 points group. And, populations with scores of 5 exhibited a markedly higher proportion of MACE at 24 and 48 months in contrast to the 0–3 points group. Furthermore, the 7-point population demonstrated significantly elevated proportions of MACE at 12, 24, and 48 months compared to the 0–3-point population, and at 24 and 48 months compared to the 4-point and 5-point populations (Fig 4B).

The ROC curve (Fig 5) assessed the predictive ability of the multifactorial model for patient prognosis. The results showed that the AUC was 0.705 for the AMI model and 0.842 for the UA model, which was greater for the multifactorial model compared to the single factor affecting patient prognosis. This indicates that the multifactorial model is better than the single factor in predicting the prognosis for patients.

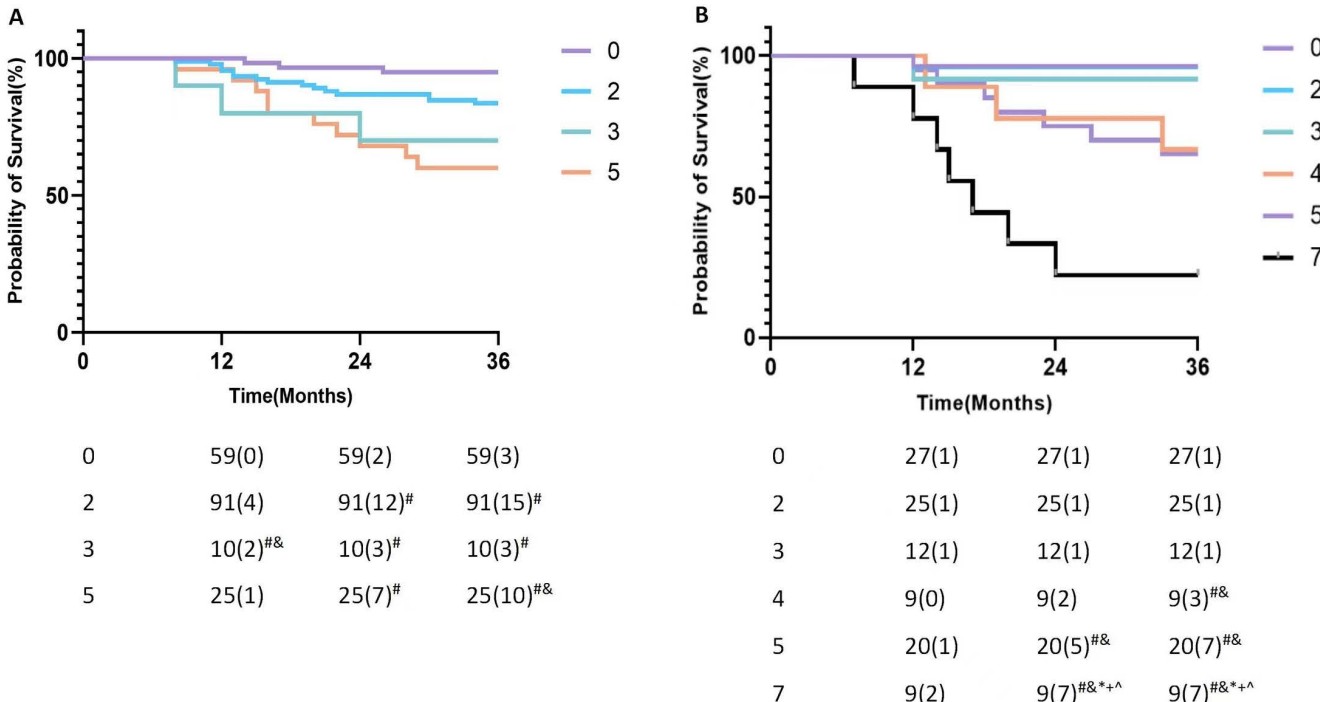

**Fig 4. The Kaplan-Meier curve of the ACS model.** A: AMI; # P<0.05 vs score=0, & P<0.05 vs score=2. B: UA; # P<0.05 vs score=0, & P<0.05 vs score=2, * P<0.05 vs score=3, + P<0.05 vs score=4, ^P<0.05 vs score=5.

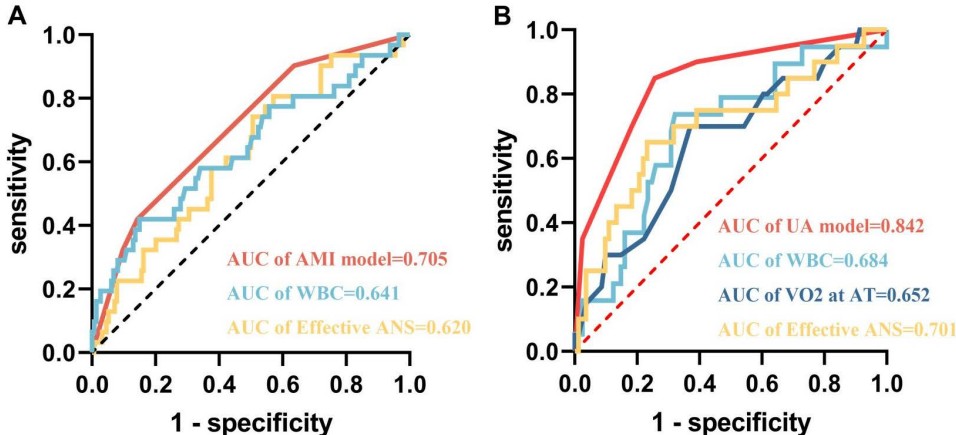

**Fig 5. The ROC curves of the AMI model and the UA model.** A: the AMI model B: the UA model.

Overall, our comprehensive prognostic risk assessment for AMI assigns a total score of 5 points, categorizing 0 points as low risk, 2 and 3 points as intermediate risk, and 5 points as high risk (Fig 6); and for UA assigns a total score of 7 points, categorizing 0–3 points as low risk, 4–5 points as intermediate risk, and 7 points as high risk (Fig 7).

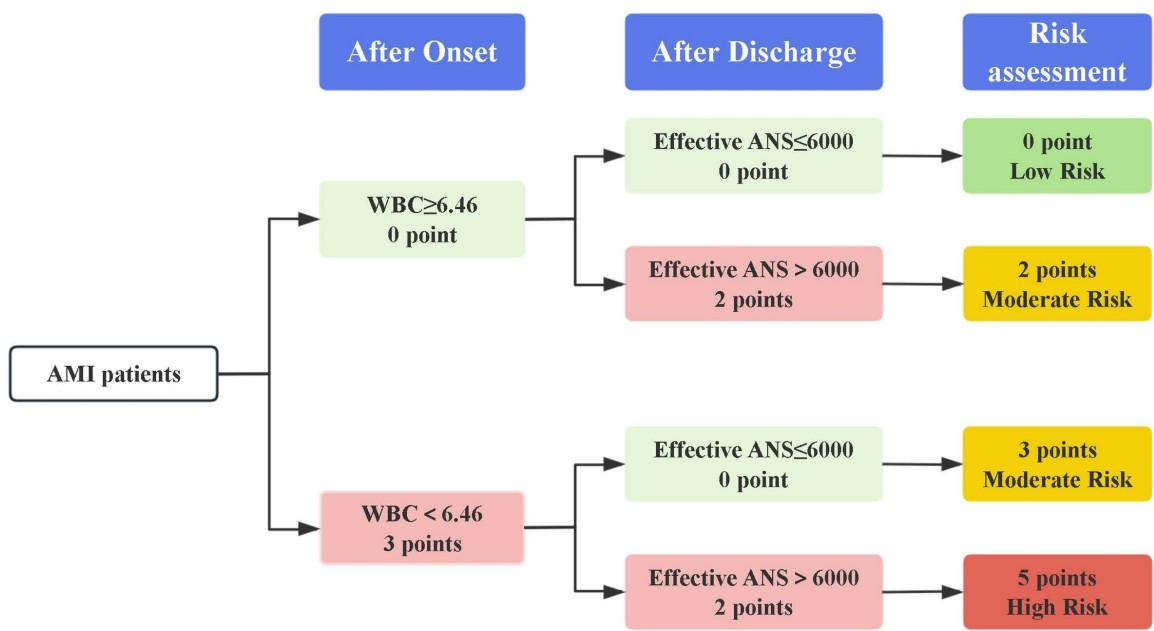

**Fig 6. Risk assessment of AMI patients by established prognostic prediction models.** WBC ($10^9$/L): white blood cell, Effective ANS (steps/day): Effective average number of steps.

## Discussion

This study retrospectively analyzed the effects of preoperative blood test indices, postoperative physical function assessment, and post-discharge daily activity on the long-term prognosis of ACS patients undergoing PCI. Prognostic prediction models were established for AMI and UA patients, respectively. The study found that the prognosis of AMI patients after PCI was related to their pre-operative WBC level and Effective ANS at home. Similarly, the prognosis of UA patients after PCI was related to their pre-operative WBC level, pre-discharge VO2 at AT, and Effective ANS at home. The risk categories for the AMI model were low (0 points), intermediate (2–3 points), and high (5 points), while for the UA model, they were low (0–3 points), intermediate (4–5 points), and high (7 points).

ACS can be classified as AMI and UA depending on the degree of myocardial damage. MI is defined as cardiomyocyte death due to prolonged ischemia, resulting in the release of cTn into the circulation [26]. In contrast, UA develops without acute myocyte injury or necrosis [27], and its pathology is distinct from that of AMI. Consistent with previous research [28], we observed a significantly higher WBC count in patients with AMI than those with UA upon presentation. During the initial stages of AMI, WBC, specifically neutrophilic granulocytes, rapidly infiltrate the infarct area [29]. Consequently, the number of WBC and neutrophilic granulocytes in AMI patients increases, which is positively correlated with the infarct area and cTnT levels, while being negatively correlated with LVEF [30]. Additionally, it was discovered that levels of ALT and AST were significantly elevated in the group of cardiomyocytes experiencing ischaemic necrosis. This may be due to hepatic perfusion dysregulation, as well as a release into the circulation after myocardial necrosis [31,32]. As a result, distinct prognostic prediction models are required for patients with AMI and UA, as their pathological states differ. Previous ACS prognostic models have seldom differentiated between AMI and UA.

Current studies on predictors of post-operative prognosis in AMI patients have primarily focused on pre-operative patient indicators, such as demographic information, hematological parameters, and cardiac ultrasound findings. Age [33], sex [34], WBC and its subtypes [28,35], dyslipidemia [36], glucose [37], and EF [38,39] are associated with prognostic

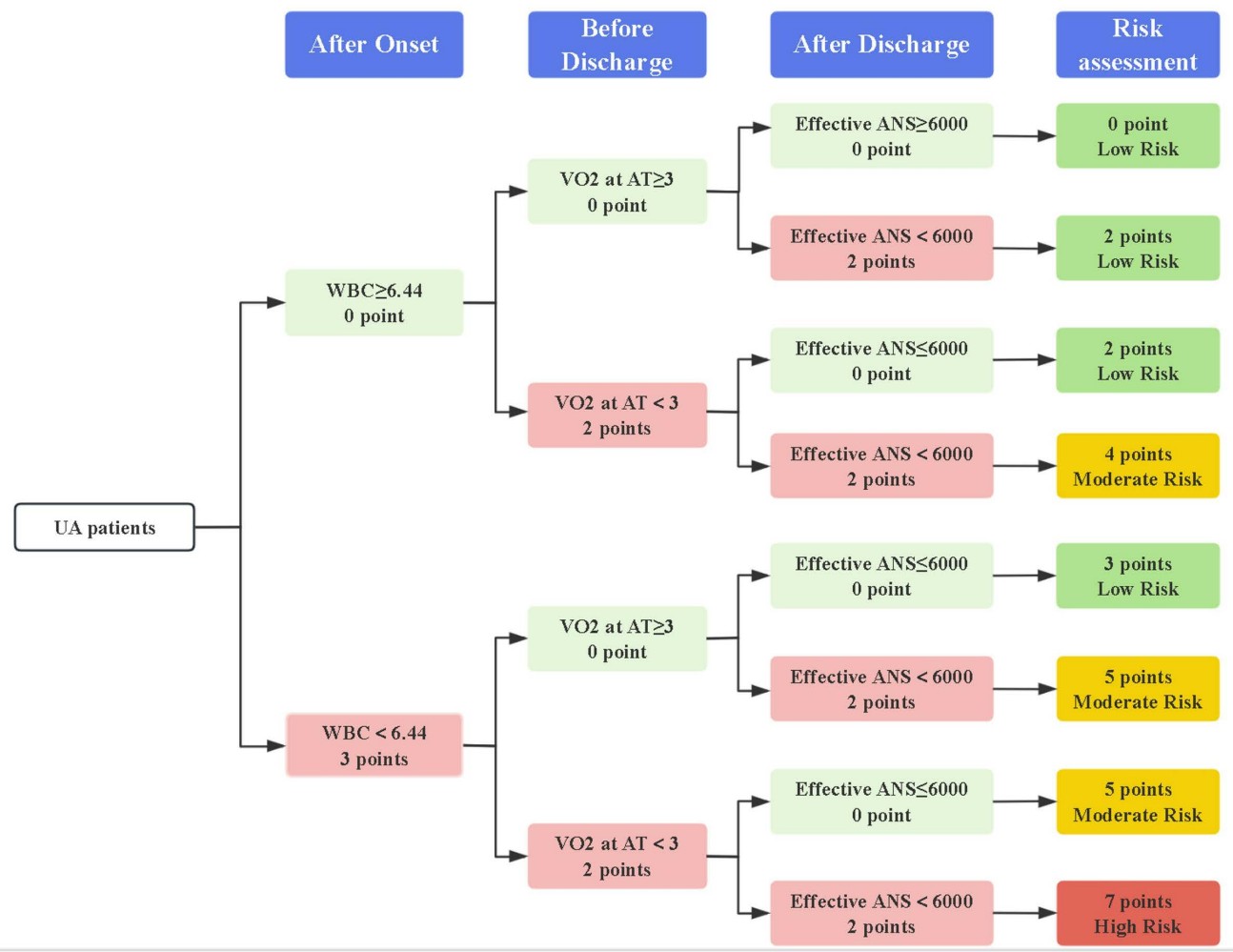

**Fig 7. Risk assessment of UA patients by established prognostic prediction models.** WBC ($10^9$/L): white blood cell, $VO_2$ at AT (Mets): Oxygen consumption per kilogram of weight per minute at anaerobic threshold, Effective ANS (steps/day): Effective average number of steps.

outcomes in AMI patients. Among the preoperative correlates included, white blood cell count (WBC) was identified as an independent risk factor for predicting adverse postoperative outcomes in patients with acute myocardial infarction (AMI). Additionally, demographic and hematological parameters, as well as related test results after presentation, are predictive of prognosis in patients with unstable angina (UA) [40–44]. Consistent with previous research on AMI, our study found that WBC count was a predictor of poor postoperative outcomes in patients with UA.

Coronary artery disease is affected by the immune system and inflammatory response [45]. Acute myocardial ischemia causes cellular injury and death in various myocardial components, including cardiomyocytes, endothelial cells, fibroblasts, and mesenchyme. This triggers a proinflammatory response through several processes, resulting in the release of various proinflammatory mediators such as cytokines, chemokines, and adhesion molecules. Following the ACS, the release of various substances, including growth factors, cytokines, and chemokines, induces the recruitment of inflammatory cells to the ischemic region of the myocardium, thereby enhancing the proinflammatory response [46]. The WBC and their subtypes are common hematological parameters that reflect the body's inflammatory state. Previous studies have disputed the correlation between post-acute WBC levels and recurrent cardiovascular events in patients with ACS.

Some studies have shown that leukocytosis is an independent risk factor and prognostic indicator of adverse cardiovascular outcomes in patients with ACS [28,35,47]. In a retrospective study performed by Cannon et al., WBC > 10.00 × 109/L was associated with increased 30 days and 10-month mortality [48]. However, in our study, a high pre-operative WBC level was considered to be a protective factor after PCI and was associated with a poor prognosis when the pre-operative WBC level was less than 6.46 × 10⁹/L in patients with AMI and less than 6.44 × 10⁹/L in patients with UA, the cut-off values we selected were within the normal range of WBC and did not fall into the category of excessively high. It has also been shown that both high and low pre-operative WBC counts predict worse outcomes in cohorts undergoing PCI [49]. Patients with lower baseline WBC counts also have higher mortality rates, and a low WBC count may be a marker of poor overall health [50].

Cardiorespiratory fitness (CRF) is widely recognized as a crucial indicator of cardiovascular health. The American Heart Association recommends CRF as the fifth most important vital sign. Low CRF has been associated with a high risk of cardiovascular disease, all-cause mortality, and mortality from various cancers, as shown by numerous studies [51]. CPET and 6MWT are both clinical tools used for assessing cardiorespiratory fitness, prescribing exercise, and predicting prognosis in patients. However, CPET requires expensive equipment and demands higher exercise capacity, while 6MWT has been proposed as a simple, inexpensive, and reproducible alternative to CPET. Both the oxygen uptake at VO2 at AT in CPET and the distance covered during the 6MWT reflect the patient's submaximal exercise capacity [52]. Recent studies have shown that CRF after PCI predicts prognosis in patients with AMI [53]. Additionally, VO2 at AT, the slope of the change in oxygen uptake to work rate (ΔVO2/ΔWR), and the slope of the ratio of minute ventilation to carbon dioxide output (VE/VCO2) have all been independently associated with MACE after PCI [13,54]. To measure postoperative CRF in patients with AMI and UA, we used different methods. For patients with AMI, we used the 6MWT, while for patients with UA, we used the ramp protocol of CPET (10 watts/min) before discharge. This allowed us to account for the varying degrees of myocardial injury in each group. It was found that 6MWD was not an independent risk factor for predicting MACE in AMI patients, whereas VO2 at AT was an independent risk factor for poor prognosis in UA patients. The reason for this may be related to the relatively lower precision of the 6MWT in assessing CRF [55], as the walking distance of the 6MWD is influenced by many factors such as gender, age, motivation, learning ability, patient exertion, and stride length [56,57]. Currently, CPET is safe in stabilized AMI patients after surgery [53], and further studies will be conducted in the future by incorporating VO2 at AT in AMI patients one month after surgery.

Daily exercise can improve the long-term prognosis of patients with ACS [14]. However, it is important to note that both excessive and insufficient exercise can have negative effects on the prognosis of patients. Recent studies suggest that the recommended number of steps for secondary prevention of cardiovascular disease is between 6,500 and 8,500 steps per day [58]. The study found that Effective ANS significantly affected the long-term prognosis of patients with AMI and UA within one year after surgery. However, the cut-off values for Effective ANS differed greatly. AMI patients with Effective ANS ≥ 6000 had significantly more MACE events in the long term, whereas UA patients with Effective ANS ≤ 6000 had fewer MACE events. This may be related to the different pathological states of AMI and UA. Necrosis of cardiomyocytes decreased cardiac function, and reduced exercise capacity are important indicators of poor prognosis in AMI patients [59]. Excessive exercise may increase the cardiac load in AMI patients and cause myocardial ischemia and hypoxia, leading to a recurrence of cardiovascular events. Therefore, exercise therapy for AMI patients needs to be more precise and individualized. On the other hand, because myocardial necrosis is absent in UA patients, a low level of daily activity may not have the desired effect of improving exercise tolerance and prognosis. Therefore, it is crucial to develop separate prognostic models for AMI and UA patients to assess prognosis.

Several simple and easily measurable biomarkers have demonstrated substantial value in the prognostic assessment of patients with acute coronary syndrome (ACS). These parameters are straightforward to calculate, cost-effective, and obtainable through routine clinical blood tests, thus offering broad applicability and significant clinical utility. By identifying high-risk patients early, they enable tailored interventions and improved clinical outcomes. The hemoglobin-to-creatinine

ratio (Hb/Cr) serves as an integrated marker of anemia and renal dysfunction, both of which contribute to chronic inflammation and hemodynamic disturbances in ACS patients, thereby influencing prognosis [60,61]. Lower Hb/Cr levels have been associated with increased all-cause mortality, higher rates of reinfarction, and a greater incidence of major bleeding at one year among patients with ACS [61]. However, the current study did not observe a significant predictive effect of Hb/Cr on post-percutaneous coronary intervention (PCI) outcomes. This discrepancy may be attributed to variations in patient characteristics, as most individuals enrolled in this study exhibited normal hemoglobin and creatinine levels, with no notable prevalence of anemia or renal dysfunction. The triglyceride-glucose (TyG) index, a surrogate marker for insulin resistance and dysregulated glucose-lipid metabolism, is closely linked to atherosclerotic progression. Prior research has identified the TyG index as not only predictive of atherosclerosis advancement but also as an independent prognostic indicator for major adverse cardiovascular events (MACE) and all-cause mortality in ACS patients [18,62]. Despite this, our study did not find a significant association between TyG and the prognosis of ACS patients following PCI. This lack of predictive value may stem from the study cohort's characteristics, where the prevalence of diabetes mellitus, dyslipidemia, and baseline triglyceride and fasting glucose levels did not significantly differ between the NO-MACE and MACE groups.

Our study found that multidimensional multifactorial constructed models can improve the limitations of single predictors. In the AMI model, WBC, although Specificity was better (Specificity = 0.850), had poorer Sensitivity (Sensitivity = 0.419) and AUC = 0.641. Effective ANS, although Sensitivity was better (Sensitivity = 0.806), Specificity was poor (Specificity = 0.429), AUC = 0.620. when the two factors were used together to construct the predictive model, both Sensitivity and AUC improved (Sensitivity = 0.903, AUC = 0.705), thus improving the accuracy of the model prediction. In the UA model, the three-factor joint prognostic model (Sensitivity = 0.850, Specificity = 0.744, AUC = 0.841), similarly improved the accuracy of the predictive model compared to a single predictor (WBC: Sensitivity = 0.737, Specificity = 0.679, AUC = 0.685; VO2 at AT: Sensitivity = 0.700, Specificity = 0.630, AUC = 0.685; Effective ANS: Sensitivity = 0.650, Specificity = 0.768, AUC = 0.701).

Scores were assigned to the multifactorial joint prediction model to improve its practicality and accuracy. The total AMI model score was 7 points, with WBC accounting for 3 points (the abnormal value was < 6.46 × 109/L) and Effective ANS accounting for 2 points (the abnormal value was > 6,000 steps/day). Both factors were normal (0 points) as low risk. One factor was abnormal (2–3 points) as intermediate risk. Both factors were abnormal (5 points) as high risk. The total UA model score was 7 points, with WBC accounting for 3 points (the abnormal value was < 6.44 × 109/L), VO2 at AT accounting for 2 points (the abnormal value was < 3Mets), and Effective ANS accounting for 2 points (the abnormal value was > 6000 steps/day). All factors were normal or only one factor was abnormal (0–3 points) as low risk, two factors were abnormal (4–5 points) as intermediate risk, and all three factors were abnormal (7 points) as high risk. The multifactorial prediction model considers different periods and factors to improve the limitation of single-factor prognosis prediction. It accurately stratifies a patient's risk, allowing for individualized, targeted, and precise rehabilitation for patients with different risk stratification [12,63]. The construction of the multidimensional multifactor model allows for secondary prevention to go beyond a single-factor indicator and achieve dynamic management of long-term prognostic risk stratification in both time and space. The proposed model offers several advantages over traditional models. Firstly, the model incorporates both the pre-discharge exercise tolerance and the post-discharge daily steps into the calculation of the MACE risk. Secondly, it allows for the stratification of risk, underscores the significance of daily activity-based cardiac rehabilitation in improving the prognosis of patients with ACS, and provides guidance on the diverse rehabilitation strategies necessitated in clinical practice for patients with AMI and UA.

There are several limitations to our current study. Firstly, all patients were recruited from a single center with a limited sample size. Further studies in larger multicenter cohorts may be necessary to validate our findings. Secondly, we only measured patients' daily activity in terms of daily steps, which did not assess activity intensity or other activity-related indicators. This requires further analysis in the future. Thirdly, we found that the 6MWT results were imprecise in assessing cardiopulmonary function in AMI patients before discharge. CPET can be performed after 1 month of stabilization of AMI

patients, thus refining the AMI model. Additionally, the patients reviewed in our study were admitted from the emergency department and underwent PCI according to the Chinese guidelines; however, the exact time from the onset of symptoms to the opening of the vessel could not be obtained from the medical records at this time. This factor may influence prognosis, we acknowledge that our study did not account for the duration of myocardial ischemia, which is a limitation.

## Conclusion

This study found that the inflammatory response after onset, physical performance and exercise tolerance before discharge, and daily activity after discharge were the independent risk factors for predicting the long-term prognosis of patients with ACS. The multidimensional prognostic model to risk-stratify for the patients with ACS, was better than the single factor model. This study also provides a theoretical basis that the prognosis of potentially high-risk patients can be improved by precise and rational exercise prescription.

## Supporting information

**S1 Dataset. Research data.**
(XLSX)

## Author contributions

**Conceptualization:** Pengyu Cao.

**Data curation:** Bojian Wang, Yanwei Du, Jinting Yang, Lijing Zhao.

**Formal analysis:** Bojian Wang, Yanwei Du, Jinting Yang, Ningning Zhang.

**Investigation:** Bojian Wang, Yanwei Du, Min Liu, Jinting Yang, Ningning Zhang.

**Methodology:** Bojian Wang, Yanwei Du, Pengyu Cao, Min Liu, Jinting Yang, Ningning Zhang.

**Resources:** Wangshu Shao, Rongyu Li, Lin Wang.

**Supervision:** Pengyu Cao, Min Liu, Lijing Zhao.

**Visualization:** Bojian Wang, Pengyu Cao.

**Writing – original draft:** Bojian Wang, Yanwei Du.

**Writing – review & editing:** Pengyu Cao.

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
