## [Decision Letter · Decision Letter 0]

16 Feb 2025

PONE-D-24-46383Development of a multidimensional prediction model for long-term prognostic risk in patients with acute coronary syndromes after percutaneous coronary intervention�a retrospective observational cohort studyPLOS ONE

Dear Dr. Cao,

Thank you for submitting your manuscript to PLOS ONE. After careful consideration, we feel that it has merit but does not fully meet PLOS ONE’s publication criteria as it currently stands. Therefore, we invite you to submit a revised version of the manuscript that addresses the points raised during the review process.

Please note that we have only been able to secure a single reviewer to assess your manuscript. We are issuing a decision on your manuscript at this point to prevent further delays in the evaluation of your manuscript. Please be aware that the editor who handles your revised manuscript might find it necessary to invite additional reviewers to assess this work once the revised manuscript is submitted. However, we will aim to proceed on the basis of this single review if possible. The reviewer has commented particularly on aspects of the study design and reporting, as well as the discussion. Please ensure you address each of the reviewer's comments when revising your manuscript. In addition, we ask that you ensure that you have adhered to the recommendations of the TRIPOD checklist in reporting the development of your predictive model: https://www.tripod-statement.org/wp-content/uploads/2020/01/Tripod-Checlist-Prediction-Model-Development.pdf

We look forward to receiving your revised manuscript.

Kind regards,

Hugh Cowley

Staff Editor

PLOS ONE

Journal Requirements:

“This assessment is funded by Science and Technology Development Plan of Jilin Province: No. 20210204119YY. The funding bodies were not involved in the study design, data collection or analysis, or writing of the manuscript.”

“This assessment is funded by Science and Technology Development Plan of Jilin Province: No. 20210204119YY. The funding bodies were not involved in the study design, data collection or analysis, or writing of the manuscript.”

Reviewers' comments:

Reviewer's Responses to Questions

**Comments to the Author**

1. Is the manuscript technically sound, and do the data support the conclusions?

Reviewer #1: Yes

2. Has the statistical analysis been performed appropriately and rigorously? 

Reviewer #1: Yes

3. Have the authors made all data underlying the findings in their manuscript fully available?

Reviewer #1: Yes

4. Is the manuscript presented in an intelligible fashion and written in standard English?

Reviewer #1: Yes

5. Review Comments to the Author

Reviewer #1: The presented study focuses on the development of a multidimensional predictive model for assessing long-term survival in patients who have undergone percutaneous coronary angioplasty following acute coronary syndromes. This is a cohort, observational study. The study included patients treated for myocardial infarction and unstable angina. The model was developed using selected laboratory and clinical parameters as well as indicators related to patients' physical activity. It appears that the ability to engage in physical activity and record its intensity determined the characteristics of the studied population and the values of EF and EDLV. The study was conducted correctly, and the statistical analysis was performed appropriately. However, the article requires several revisions and clarifications.

1. The authors must decide whether they intend to use the term fasting blood sugar or fasting plasma glucose, as both are used interchangeably throughout the article.

2. The authors must specify whether the reported values of laboratory and echocardiographic parameters were measured at admission, at discharge, or averaged.

o The EF values in AMI are surprisingly high: EF (%), (IQR) 57.00 (55.00,60.00) 58.00 (53.00,60.00).

3. The presented study has several limitations that require description: a. The variables did not include troponin levels. b. The duration of ischemia was not considered. c. The number of affected coronary arteries was not included. d. The type of myocardial infarction was not specified, i.e., whether it was an ST-elevation myocardial infarction (STEMI) or a non-ST-elevation myocardial infarction (NSTEMI). e. The study was conducted, and consequently, the model was developed based on a relatively small population. f. The hemodynamic status of patients at hospital admission (Killip-Kimball classification) was not considered.

4. It is recommended to expand the discussion section with references to other studies describing the significance of laboratory indicators in cardiovascular risk assessment, for example: a. https://doi.org/10.1016/j.ijcard.2024.132663. b. https://doi.org/10.1093/eurheartj/ehae666.1571.

5. Considering the descriptive characteristics of the AMI group, the authors should reconsider modifying the title and approach to the model as a predictive model for assessing long-term prognosis in patients with preserved ejection fraction.

6. PLOS authors have the option to publish the peer review history of their article (what does this mean? ). If published, this will include your full peer review and any attached files.

**Do you want your identity to be public for this peer review?** For information about this choice, including consent withdrawal, please see our Privacy Policy .

Reviewer #1: No

---

## [Author Response · Author response to Decision Letter 1]

16 Mar 2025

The point-by-point responses to the criticisms and suggestions are presented in "Response to Reviewer" file.

---

## [Decision Letter · Decision Letter 1]

27 Mar 2025

Development of a multidimensional prediction model for long-term prognostic risk in patients with acute coronary syndromes after percutaneous coronary intervention�a retrospective observational cohort study

PONE-D-24-46383R1

Dear Dr. Cao,

We’re pleased to inform you that your manuscript has been judged scientifically suitable for publication and will be formally accepted for publication once it meets all outstanding technical requirements.

Kind regards,

Giuseppe Andò, M.D., Ph.D.

Academic Editor

PLOS ONE

Additional Editor Comments (optional):

Reviewers' comments:

Reviewer's Responses to Questions

**Comments to the Author**

1. If the authors have adequately addressed your comments raised in a previous round of review and you feel that this manuscript is now acceptable for publication, you may indicate that here to bypass the “Comments to the Author” section, enter your conflict of interest statement in the “Confidential to Editor” section, and submit your "Accept" recommendation.

Reviewer #1: All comments have been addressed

2. Is the manuscript technically sound, and do the data support the conclusions?

Reviewer #1: Yes

3. Has the statistical analysis been performed appropriately and rigorously? 

Reviewer #1: Yes

4. Have the authors made all data underlying the findings in their manuscript fully available?

Reviewer #1: Yes

5. Is the manuscript presented in an intelligible fashion and written in standard English?

Reviewer #1: Yes

6. Review Comments to the Author

Reviewer #1: The manuscript has been revised according to the reviewer's suggestions. This predictive model should be verified with a large sample. A variable such as ejection fraction should play an important role in long-term prognosis after ACS.

7. PLOS authors have the option to publish the peer review history of their article (what does this mean? ). If published, this will include your full peer review and any attached files.

**Do you want your identity to be public for this peer review?** For information about this choice, including consent withdrawal, please see our Privacy Policy .

Reviewer #1: No

---

## [Editor Report · Acceptance letter]

PONE-D-24-46383R1

PLOS ONE

Dear Dr. Cao,

I'm pleased to inform you that your manuscript has been deemed suitable for publication in PLOS ONE. Congratulations! Your manuscript is now being handed over to our production team.

Kind regards,

on behalf of

Prof. Giuseppe Andò

Academic Editor

PLOS ONE